# Post-Traumatic Stress Disorder and Mental Health in Chinese College Athletes during and after the COVID-19 Pandemic: The Multiple Mediating Effects of Basic Psychological Needs and Generalized Anxiety Disorder

**DOI:** 10.3390/bs13070567

**Published:** 2023-07-07

**Authors:** Xiuhan Zhao, Zongyu Liu, Liangyu Zhao, Qi Wang, Liguo Zhang

**Affiliations:** School of Physical Education, Shandong University, Jinan 250061, China; zhaoxiuhan@sdu.edu.cn (X.Z.); 201700292037@mail.sdu.edu.cn (Z.L.); zhaoliangyu@mail.sdu.edu.cn (L.Z.); 202135355@mail.sdu.edu.cn (Q.W.)

**Keywords:** post-traumatic stress disorder, mental health, basic psychological needs, generalized anxiety disorder

## Abstract

Psychological issues have a complex and multidimensional impact on a college athlete’s training and performance. As the reserve talent of competitive sports, it is very important to investigate the psychological health of athletes. This study aims to explore the association between generalized anxiety disorder (GAD), post-traumatic stress disorder (PTSD), basic psychological needs (BPN), and mental health (MH) among Chinese college athletes. Chinese college athletes who were willing to participate in the study, had participated in national competitions, and had a Chinese athlete rating certificate were included in this study. Participants completed the Kessler Psychological Distress Scale (K10), the Generalized Anxiety Disorder Scale (GAD-7), the Impact of Event Scale-Revised (IES-R), and the Basic Needs Satisfaction in General Scale (BNSG-S). The study involved 665 college athletes (415 males and 250 females), with an average age of 20.43 years (SD = 1.68). We performed descriptive statistics, correlation analyses, and moderated chain mediation analyses using SPSS 22.0 and Hayes’ PROCESS macro. The results of the final model showed that basic psychological needs were positively correlated with mental health (*r* = 0.443, *p* < 0.001), while PTSD (*r* = −0.346, *p* < 0.001) and generalized anxiety disorder (*r* = −0.527, *p* < 0.001) were negatively correlated with mental health among college athletes. There were significant indirect impacts. According to the bootstrapping results, basic psychological needs and generalized anxiety disorder played a mediating role in 22.54% and 50.29% of the total effects of PTSD on mental health, respectively. Meanwhile, the chain mediating effect of basic psychological needs and generalized anxiety disorder (7.23%) was also significant. The study’s findings advance our understanding of the connections between PTSD and mental health and highlight the significant roles played by basic psychological needs and generalized anxiety disorder in that link among Chinese college athletes.

## 1. Introduction

There is evidence that college students have a higher prevalence of mental health illnesses than their non-college peers, which is a serious public health concern that has caught society’s attention recently [1,2]. It may be harmful to a college athlete’s physical and emotional development to intensify training or competition during this special period of growth [3]. College athletes’ mental health needs are of special concern because they can affect how well they train, according to Moreland et al. [4]. This group’s mental health problems need to be addressed in addition to their physical abilities [5]. There is no question that the COVID-19 pandemic has had a negative psychological impact on some college students, particularly athletes [6], which could have a negative impact on competition outcomes and even the country’s supply of top talent. As a result, it is crucial to face up to athletes’ mental health issues and choose the best course of action to improve them.

### 1.1. Post-Traumatic Stress Disorder and Mental Health

PTSD is defined in the current study as a distinct and negative emotional state brought on by unexpected changes in the external environment (COVID-19 pandemic) [7], which is the outcome of cognition and the evaluation of external resources. Researchers have discovered a link between pandemic-related stressors like contagion risk, post-traumatic stress disorder, future uncertainty, worry, and mental health issues. It has been shown that psychological stress can have long-term effects on the body’s physical and mental health [8]. In order to protect athletes’ health, several sporting events have been postponed or canceled, which has a detrimental psychological effect on the athletes [7]. According to research, some groups may be more sensitive than others to the COVID-19 pandemic’s psychological impacts [9]. Research suggests that young people’s mental health has become a global concern, and the subjective discomfort caused by the COVID-19 pandemic has had a major negative impact on their mental health [10]. Numerous studies [11,12,13] have examined how the COVID-19 epidemic has affected the mental health of frontline healthcare providers, COVID-19 patients, the general population, and university students. These studies have demonstrated that both confirmed and suspected COVID-19 patients exhibit varying degrees of psychological stress symptoms, with some even developing post-traumatic stress disorder (PTSD). Aron et al. have found that elite athletes may exhibit higher rates of PTSD and other trauma-related disorders and that symptoms of PTSD may significantly damage athletes’ mental health [14]. Another study found that a post-traumatic psychological intervention approach could enhance respondents’ psychological flexibility for mental health recovery in traumatized athletes [15]. College students (including college athletes) are currently in a critical stage of transition from adolescence to adulthood and healthy physical and mental development [16]. According to a previous study, college students are more vulnerable to various effects of sudden changes in the COVID-19 pandemic, and these persistent COVID-19 pandemic-related stressors may adversely affect their mental health [17].

### 1.2. The Mediating Role of Basic Psychological Needs

Basic psychological needs [18] refer to people’s innate, intrinsic, and necessary psychological needs that are essential to the healthy development, integration, and happiness of individual psychology. These needs include those for competence, relatedness, and autonomy [19]. According to the self-determination theory, basic psychological needs are the driving forces behind environmental factors that have an impact on individual behavior and are necessary for healthy individual development and the realization of optimal functions [20]. When facing the COVID-19 epidemic, it is undoubtedly difficult for individuals to maintain fundamental psychological demands. The uncertainty of competition or the risk of infection during the COVID-19 pandemic may cause athletes to experience post-traumatic stress symptoms, which may reduce their autonomy, increase their frustration, cut off their connection to some social relations, and affect their satisfaction of competence needs [21]. According to one study [22], meeting basic psychological needs can significantly lower sadness, anxiety, and other psychological issues, as well as positively predict happiness and mental health. Bandura’s cognitive appraisal theory suggests that an individual’s perceived threat after experiencing a negative event affects the individual’s need for competence satisfaction, which in turn can cause more negative emotions [23]. Hodge and Lonsdale et al. found that the basic psychological needs of professional athletes could significantly and negatively predict their psychological fatigue [24,25]. Although there is a lack of evidence regarding the relationship between PTSD and basic psychological needs in professional or amateur athletes, this relationship has been validated in youth populations [26].

### 1.3. The Mediating Role of Generalized Anxiety Disorder

The hallmark of generalized anxiety disorder, which is a common chronic mental illness among college students, is excessive, hard-to-control anxiety that causes considerable suffering and impairment [27]. Post-traumatic stress symptoms have been linked to anxiety disorders in earlier studies. Because of their unique circumstances and condensed training and competition schedules, college athletes are more likely to have anxiety symptoms [28]. Due to the COVID-19 pandemic, college athletes’ living and training spaces are even more constrained. The living and training areas of athletes have been somewhat impacted by the epidemic’s spread. Their capacity to control their mind and body has recently been tested, and they are more vulnerable to psychological negativity like generalized anxiety disorder or other conditions that are very bad for their mental health. Guevarra et al. found that the sudden outbreak of the COVID-19 epidemic and the special situation of sports competitions put athletes under a high level of stress and tension, which led to the development of anxiety disorders in athletes [29]. Recent studies have shown that college athletes are prone to anxiety disorders, which can impair their mental health status [30,31]. Additionally, Malekinezhad et al.’s [32] study discovered that risk factors for positive mental health, stress-related mental diseases, and anxiety were related to each individual’s mental health.

### 1.4. The Present Study

In summary, athletes’ mental health during and after the COVID-19 pandemic and post-pandemic period is a cause for concern. Many studies have verified the negative impact of PTSD on individuals’ mental health and the impact of basic psychological needs and generalized anxiety disorder on promoting mental health status. However, the association between PTSD and athletes’ mental health during and after the COVID-19 pandemic and the psychological mechanisms underlying this association have been relatively understudied in previous research (especially in the Chinese context), with the majority of research concentrating on non-athletic college students. To fill this research gap, this study conducted an in-depth investigation of the relationship between PTSD and mental health in a group of Chinese college athletes in the context of the current gradual reopening of competitive matches. Specifically, it is an innovation of this study to introduce basic psychological needs and generalized anxiety disorder as mediating variables into the influence mechanism of PTSD on mental health among Chinese college athletes, which makes up for the deficiency of current research to some extent. The results of this study may provide evidence-based interventions for this group and have practical value in improving the mental health of college athletes, improving their athletic performance, and providing evidence for national talent selection. Based on the significant correlation between these factors, we developed a chain mediation model (Figure 1) in this work to investigate the connections among PTSD, basic psychological needs, generalized anxiety disorder, and mental health in athletes in a Chinese cultural setting. Figure 1 illustrates three mediation paths: β1–β6, β5–β3, and β1–β2–β3. In order to prepare for the following stage of the study, correlation analysis was first utilized to explore the relationships between the variables. The impacts of PTSD on mental health were investigated using chain-mediated effects analysis to determine whether basic psychological needs and generalized anxiety disorder were mediating factors.

## 2. Methods

### 2.1. Study Design

The study, which used a cross-sectional design, took place in China in February 2022 during the COVID-19 epidemic, and Chinese college athletes who were willing to participate in the study, had participated in national competitions, and had a Chinese athlete rating certificate were asked to take part. The study adheres to the STROBE methodology for cross-sectional studies [33].

### 2.2. Setting and Participants

From the 72 teams in the Chinese college soccer league, 35 teams were chosen at random. These college athletes come from 29 provinces and administrative regions in China, including Beijing, Shanghai, Chongqing, etc. Before sending an electronic questionnaire to all athletes via a link to the QR code, the coaches of each team first explained the study’s goals and methods to them. They also notified them of their right to withdraw from the study at any time. Each participant provided informed consent online before beginning the study. The study included 694 athletes in all. Questionnaires that were incomplete and had the same answer for all questions were identified as invalid and excluded. A total of 665 subjects were included after excluding invalid questionnaires, with an effective recovery rate of 95.8%.

### 2.3. Questionnaires

Age, sex, urban-rural provenance, and length of sports careers were among the sociodemographic factors that were gathered through the use of sociodemographic questions.

Using the Chinese version of the Impact of Event Scale-Revised (IES-R) [34], this study evaluated the PTSD experienced by college athletes caused by the COVID-19 epidemic. The scale, which had 22 items and a Likert scale with a range from 0 to 4 for each, was available to participants over the age of 17. A person’s response to recent life events that may have exposed them to severe post-traumatic stress symptoms in the preceding week was evaluated. The overall score, which ranges from 0 to 88, indicates how distressed people are as a result of the COVID-19 epidemic. This measure has been used in a prior study to monitor Italian athletes’ perceptions of psychological distress during the COVID-19 outbreak, and it has been shown to have good reliability and validity [7]. The scale’s Cronbach’s alpha coefficient in this study was 0.952.

The Chinese version of the Generalized Anxiety Disorder 7 Scale (GAD-7) is a concise and rigorously validated scale of anxiety symptoms [35], which has been proven to be diagnostic enough to examine a wide range of anxiety-related disorders (generalized anxiety disorder, panic disorder, and social phobia). A 4-point Likert scale (0–3) was used to assess seven questions in order to take into account the frequency of anxiety symptoms during the previous seven days. The scale, which is useful for evaluating anxiety in college students, has a total score that ranges from 0 to 21, with the severity of the anxiety increasing as the score rises. The scale’s Cronbach’s alpha coefficient for this study was 0.933.

The Chinese version of the Basic Needs Satisfaction in General Scale (BNSG-S), which was translated and revised by Yu et al. [36], includes the 21 items from the original scale as well as the three dimensions of autonomy need, competence need, and relatedness need, and is used to measure how well each person’s basic psychological needs are met. The Likert scale, which ranges from “completely inconsistent” to “completely consistent,” is used. The degree to which a basic psychological need is satisfied increases with the total score. The scale’s (BNSG-S) validity and reliability have been verified. The scale’s Cronbach’s alpha coefficient for this investigation was 0.864.

The Chinese version of the Kessler Psychological Distress Scale (K10) [37] contained 10 items on the frequency of symptoms associated with generalized mental health disorders, such as anxiety and stress levels, experienced over the previous month. A 5-point scale (1 = almost none, 2 = occasionally, 3 = some of the time, 4 = most of the time, and 5 = all the time) was used, with the higher the score, the worse the mental health status. The earlier study demonstrated that the scale had respectable validity and reliability. The scale’s Cronbach’s alpha coefficient for this investigation was 0.916.

### 2.4. Statistical Analysis

The original data were exported from the Wenjuanxing questionnaire platform (https://www.wjx.cn, accessed on 1 January 2022). All the statistical analyses were performed with SPSS version 22.0 (IBM, Armonk, NY, USA) for Windows. The association between PTSD, mental health, basic psychological needs, and generalized anxiety disorder in Chinese college athletes was examined using Pearson’s correlation coefficients. To examine the direct association between PTSD and mental health as well as the mediating effect of basic psychological needs and generalized anxiety disorder, Hayes’ PROCESS macro in SPSS (version 3.3) was used [38].

### 2.5. Ethical Considerations

The Ethics Committee of Shandong University granted approval for this study (No. 2021-1-114).

## 3. Results

### 3.1. Common Method Bias Testing

Considering that the results may be impacted by common method bias because the data were collected using a self-reported approach, an exploratory factor analysis was performed on each item using Harman’s single-factor test approach. The variance explanation rate of the first factor in this study was 28.23%, which is much lower than the judgment threshold of 40% and suggests that common method bias had less of an impact on the data gathered through a questionnaire survey.

### 3.2. Descriptive Statistics and Correlation Analysis

The demographics of the final sample can be observed in Table 1. The results of the correlation analysis of the variables showed that the four variables were significantly correlated to each other (Table 2).

Table 1 shows that participants’ mean age (standard deviation = 1.68) was 20.43. The majority of athletes (45.5%) are between the ages of 19 and 20. The athletes’ gender split was 62.4% female and 37.6% male. While 45.3% of people were raised in rural areas, 54.7% were not. Additionally, the biggest percentage (59.2%) was made up of athletes whose sporting careers lasted 6–10 years. 

Table 2′s correlation results show that PTSD was positively connected with generalized anxiety disorder (*r* = 0.505, *p* < 0.001) and negatively correlated with basic psychological needs (*r* = −0.272, *p* < 0.001) and mental health (*r* = −0.346, *p* < 0.001). Basic psychological needs were positively connected with mental health (*r* = 0.443, *p* < 0.001) and negatively correlated with generalized anxiety disorder (*r* = −0.527, *p* < 0.001), respectively. Additionally, there was a negative correlation between basic psychological demands and generalized anxiety disorder (*r* = −0.353, *p* < 0.001).

### 3.3. Chain Mediating Model

This study tested the chain mediation model using Model 6 in the Process 3.3 macro of SPSS 22.0, where the underlying demographic variables were used as control variables. The results showed that basic psychological needs and generalized disorders fully mediated the effect of PTSD on the mental health of Chinese college athletes, as shown in Table 3 and Table 4 and Figure 2.

Results (Table 3) showed that PRSD was a negative predictor of MH (β = −0.346, *p* < 0.001). When BPN and GAD were introduced as mediating variables, the predictive power of PRSD on MH was not significant (β = −0.069, *p* > 0.05). PRSD had a significant predictive effect for BPN (β = −0.271, *p* < 0.001) and GAD (β = 0.441, *p* < 0.001). In addition, BPN significantly predicted individual GAD (β = −0.230, *p* < 0.001).

According to the results of the test for the mediating effect of basic psychological needs (Table 4), basic psychological needs mediated the effect of PTSD on mental health, with the mediating effect being 0.078, accounting for 22.54% of the total effect. The upper and lower limits of the bootstrap 95% confidence interval did not contain 0, indicating that basic psychological needs mediated the effect of PTSD on mental health. According to the findings of the effect of generalized anxiety disorder mediation, generalized anxiety disorder mediated the impact of PTSD on mental health with a mediating effect of 0.174, or 50.29% of the total effect. According to the results of the test of the chain-mediating effect of basic psychological needs and generalized anxiety disorder, the chain-mediating effect was substantial, with a mediating effect of 0.025, accounting for 7.23% of the total effect. 

As illustrated in Figure 2, PTSD was not significantly related to mental health after introducing the chain mediating variables of basic psychological needs and generalized anxiety disorder, indicating that basic psychological needs and generalized anxiety disorder fully mediated the effect of PTSD on the mental health of Chinese college athletes.

## 4. Discussion

At present, the development and talent delivery of Chinese athletes have aroused widespread concern in society. Therefore, it’s important to pay attention to college athletes. The COVID-19 pandemic had a negative influence on their mental health, which needs to be fully studied given its importance to their growth. This study used a questionnaire survey to investigate the connection between PTSD and mental health. The results of the current study showed that PTSD was negatively connected with mental health and that the association between PTSD and mental health was indirectly mediated by basic psychological needs and generalized anxiety disorder. 

Our findings confirmed earlier research by finding a significant negative connection between PTSD and MH. The COVID-19 outbreak has negatively impacted people’s daily lives, places of employment, and academic pursuits, resulting in a variety of psychological reactions [6,8]. The interaction of the organism with the environment that leads to PTSD is a comprehensive response to stimulation events that disrupt the organism’s homeostasis and surpass the load and control of the organism. College athletes are still going through the process of developing their minds; as a result, their mental faculties are still developing and are not mature enough to make well-informed decisions. College athletes have to face various challenges in training and competition, such as competition pressure, injuries in training, interpersonal relationships within sports teams, etc. [39]. When faced with a particular situation, athletes cannot help but display a variety of physiological and psychological reactions (breathlessness, accelerated heartbeat, mental strain, etc.). In addition to physical illness, they may also experience mental health issues as a result of the internal environment being destroyed by prolonged, continuous overload stress episodes [40].

Additionally, this study revealed that basic psychological needs play a mediation role in the relationship between PTSD and mental health, which supports the self-determinism theory [41]. A person’s ability to meet their basic psychological requirements depends on the environment’s capacity to offer adequate supportive resources. The satisfaction of athletes’ basic psychological needs, which are the essential nutrients required for the development of individual mental health, is somewhat impacted by the PTSD brought on by COVID-19. The level of each individual’s mental health will suffer as a result [42]. Additionally, we discovered that generalized anxiety disorder mediates the link between PTSD and mental health in college athletes. During the COVID-19 pandemic, it was discovered that college students had significant psychological issues, such as generalized anxiety disorder [43]. Since their mental health is impacted by their special role, career advancement, and other factors, college athletes and students are more vulnerable to physical and mental reactions and experience more negative emotions under the COVID-19 pandemic.

Another significant finding of this study is the role of generalized anxiety disorder and basic psychological needs in mediating the relationship between PTSD and mental health. People who had PTSD as a result of the COVID-19 pandemic may have perceived less social support. It is therefore challenging for people to satisfy their basic psychological needs, which may result in generalized anxiety disorder and ultimately have an impact on their mental health. According to the Self-Determination Theory, a positive social environment can help an individual feel competent and autonomous and can assist in meeting basic psychological requirements (e.g., subjective support, objective support, etc.) [44]. During the COVID-19 pandemic, people were advised to avoid intimate contact since it reduces a person’s sense of social support and emotions and impairs their satisfaction of their need for relatedness [45]. According to self-determination theory [46], people’s feelings of competence will rise and satisfy their need for competence when they believe they have more interpersonal resources at their disposal and better coping skills. Therefore, the restricted formation of social relationships due to PTSD cannot satisfy the desire for relatedness. People may feel more liberated to act independently and pursue their own objectives when they believe that everyone is on their side. Athletes are being isolated in specific locations to practice due to the coronavirus pandemic, which prevents them from leaving and socializing as usual, which exacerbates their generalized anxiety disorder. Their generalized anxiety disorder regarding the possibility of stressful occurrences can be lessened if they believe they have a large social network [47]. Higher levels of psychological need satisfaction were linked to lower levels of depression and generalized anxiety disorder, according to a meta-analysis [48]. Generalized anxiety disorder levels decrease, and mental health is enhanced when basic psychological needs are satisfied [49,50].

This research has important theoretical value and practical significance. Theoretically, it is important to comprehend how PTSD affects Chinese athletes’ mental health in order to further theorize about the process underlying their relationship. In addition, it offers the self-determination theory some empirical support. According to the internal mechanism of PTSD impacting individual mental health, it is advantageous to bring forward more targeted methods to support the mental health growth of Chinese college athletes, which may enhance their athletic performance. College administrators should pay special attention to the influence of information dissemination on the change of college students’ mentality and carry out scientific mental health education and intervention. Support from the schools will also aid in reducing PTSD among college students. In order to help college students deal with anxiety and difficult emotional reactions when confronted with this public health issue, colleges and universities might build a comprehensive and varied psychological support system.

From the viewpoint of positive psychology, this study investigates the environmental and individual factors that affect the mental health of Chinese college athletes. It also discusses the potential impact of basic psychological needs and generalized anxiety disorder on these athletes’ mental health. The findings of this study suggest that families, schools, and communities can help athletes who were negatively impacted by the public health crisis by meeting their basic psychological needs and lowering their generalized anxiety disorder. This will help promote their mental health and is instructive for educating college athletes. This study has some limitations. Firstly, this was a cross-sectional study, which cannot establish causality between the variables. Therefore, longitudinal studies should be conducted in the future to investigate the internal mechanisms of the factors that influence the mental health of college athletes. Moreover, this study relied on self-reported data. Additional measures that provide objective indicators of an athlete’s mental state should be included in future studies. Thirdly, the athletes in this study were all from mainland China, which may limit the generalizability of the findings.

## 5. Conclusions

In conclusion, this is the first study to examine the connection between PTSD and Chinese athletes’ mental health as well as the underlying psychological mechanisms during the public health crisis. Mental health is connected to PTSD either directly or indirectly. In particular, the chain mediating effects of generalized anxiety disorder and basic psychological needs influence the relationship between PTSD and mental health in college athletes. Interventions for PTSD that aim to improve mental health can be gained from methods that lessen generalized anxiety disorder and satisfy basic psychological needs. Future longitudinal studies, particularly those conducted in developing countries, may help clarify the psychological processes by which PTSD affects mental health.

## Figures and Tables

**Figure 1 behavsci-13-00567-f001:**
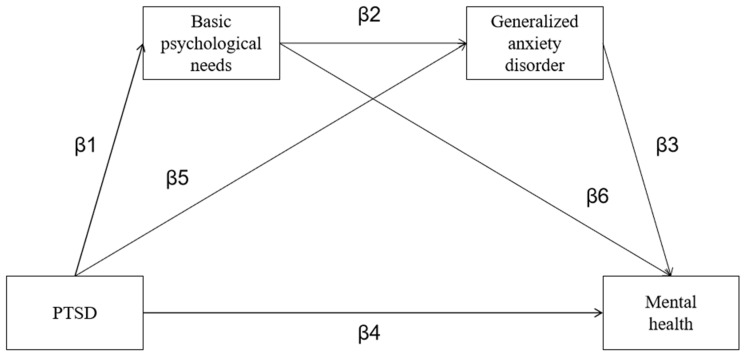
The hypothetical chain mediating effect model of basic psychological needs and generalized anxiety disorder between PTSD and mental health among college athletes.

**Figure 2 behavsci-13-00567-f002:**
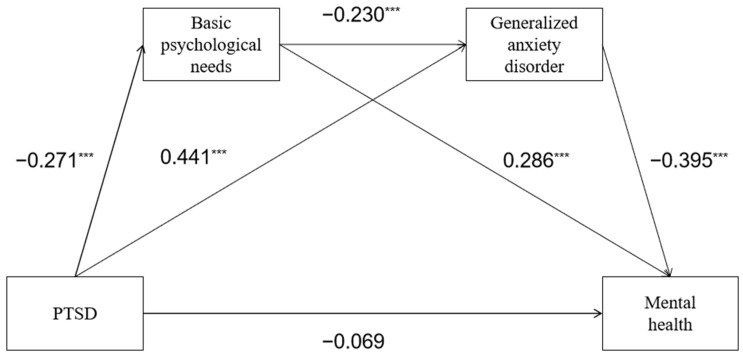
Mediating role of basic psychological needs and generalized anxiety disorder. *** *p* < 0.001.

**Table 1 behavsci-13-00567-t001:** Characteristics of athlete participants (*n* = 665).

Characteristics	Category	*n*	%
Sex *	Male	415	62.4
	Female	250	37.6
Age (years) *	17–18 years	75	11.3
	19–20 years	305	45.9
	21–22 years	207	31.1
	23–24 years	64	9.6
	25–27 years	14	2.1
Urban−rural provenance *, *n* (%)	Urban	364	54.7
	Rural	301	45.3
Duration of sports careers *	1–5 years	126	18.9
	6–10 years	393	59.2
	11–13 years	114	17.1
	14 years or more	32	4.8

Note: * Dummy variables. The data were described as *n* (%). Average age = 20.43 years (SD = 1.68).

**Table 2 behavsci-13-00567-t002:** Descriptive statistics and correlations of scale composite scores (*n* = 665).

	M (SD)	1	2	3	4
1. PTSD	1.77 ± 0.61	1			
2. Basic psychological needs	4.68 ± 0.79	−0.272 ***	1		
3. Generalized anxiety disorder	1.41 ± 0.51	0.505 ***	−0.353 ***	1	
4. Mental health	4.00 ± 0.91	−0.346 ***	0.443 ***	−0.527 ***	1

Notes: *** *p* < 0.001. M = mean, and SD = standard deviation.

**Table 3 behavsci-13-00567-t003:** The chain mediating effect of BPN and GAD (*n* = 665).

Regression Equation		Overall Fit Indices			Significance of the Regression Coefficients	
		*R*	*R* ^2^	*F*	*β*	*t*
Outcome Variables	Predictors					
Mental health	PTSD	0.371	0.137	20.965 ***	−0.346	−9.544 ***
Basic psychological needs	PTSD	0.307	0.094	13.694 ***	−0.271	−7.321 ***
Generalized anxiety disorder	PTSD	0.559	0.312	49.776 ***	0.441	13.108 ***
	Basic psychological needs				−0.230	6.773 ***
Mental health	PTSD	0.611	0.373	55.791 ***	−0.069	−1.918
	Basic psychological needs				0.286	8.512 ***
	Generalized anxiety disorder				−0.395	−10.593 ***

Notes: ****p* < 0.001.

**Table 4 behavsci-13-00567-t004:** The direct, indirect, and total effects of the chain mediation model (*n* = 665).

Effect Types	B	SE	95% CI
Total effect	−0.346	0.036	(−0.417, −0.275)
Direct effect	−0.069	0.036	(−0.140, −0.002)
Indirect effect of basic psychological needs	−0.078	0.015	(−0.108, −0.050)
Indirect effect of basic generalized anxiety disorder	−0.174	0.023	(−0.221, −0.129)
Chain mediating effects of basic psychological needs and generalized anxiety disorder	−0.025	0.005	(−0.036, −0.015)

## Data Availability

Data is available on request from the corresponding author.

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
