# Peer review of "Post-Traumatic Stress Disorder and Mental Health in Chinese College Athletes during and after the COVID-19 Pandemic: The Multiple Mediating Effects of Basic Psychological Needs and Generalized Anxiety Disorder"

_behavsci, 2023, doi:10.3390/bs13070567_

Round 1

Reviewer 1 Report

This study aims to investigate the association between 10 generalized anxiety disorder (GAD), posttraumatic stress disorder (PTSD), basic psychological 11 needs (BPN), and mental health (MH) among Chinese sportsmen.

Major comments:

- Indicate the authors' affiliation

Introduction:

- To highlight the novelty of the study

Methods:

Results:

why is the age and duration of the sports career not given in mean and standard deviation?

Discussion:

Mention the limitations of the study

Author Response

Response to Reviewer 1 Comments

Manuscript ID: behavsci-2452211

Title: Posttraumatic Stress Disorder and Mental Health in Athletes: The Multiple Mediating Effects of Basic Psychological Needs and Generalized Anxiety Disorder

Dear Reviewer,

Thank you for the opportunity to improve our manuscript. Those comments are all valuable and very helpful for revising and improving our manuscript, as well as the important guiding significance to our research. We have studied these comments carefully and have made corrections which we hope meet with approval. We have responded to each comment and the revisions are marked in red in the manuscript. We are grateful for your positive and constructive comments and the main corrections in the manuscript and the reply to the reviewer’ s comments are as flows:

-This study aims to investigate the association between 10 generalized anxiety disorder (GAD), posttraumatic stress disorder (PTSD), basic psychological 11 needs (BPN), and mental health (MH) among Chinese sportsmen.

Response: We appreciate your comments.

Point 1:

Major comments:

- Indicate the authors' affiliation

Response: We thank the reviewers suggestion. We are sorry that we did not indicate the authors' affiliation. We have now indicated the author's affiliation in the manuscript, all authors come from School of Physical Education, Shandong University, Jinan 250061, China. Please see p.1, lines 6-7,thank you.

Point 2:

Introduction:

- To highlight the novelty of the study

Response: Thank the reviewer for your advice. It is an innovation of this study to introduce basic psychological needs and generalized anxiety disorder as mediating variables into the influence mechanism of PTSD on mental health among Chinese college athletes, which makes up for the deficiency of current research to some extent. We have added the research innovation in the manuscript, please check, thank you (p.3, lines 123-126).

Point 3:

Methods:

Results:

- why is the age and duration of the sports career not given in mean and standard deviation?

Response: Thank you very much for your reminder. In our study, participants mean age (standard deviation = 1.68) was 20.43. We have mentioned this on p.6, line 223. However, in this study, we take the duration of the sports career as a dummy variable. They are divided into four categories: 1-5 years; 6-10 years; 11-13 years; 14 years or more. Instead of specifying the length of their athletic career, participants were asked to select one of four options when answering the questionnaire. In addition, we presented age as a categorical variable/dummy variable in Table 1 because we wanted to observe the proportion of athletes of all ages. Therefore, in the table 1, we emphasize Sex, Age, Urban−rural provenance and Duration of sports careers as dummy variables in the manuscript for the convenience of readers (please see table 1 and p.6, line 222).

Point 4:

Discussion:

- Mention the limitations of the study

Response: Thank you for reading our manuscript carefully. This study has some limitations. Firstly, this was a cross-sectional study, which can not establish causality between the variables. Therefore, longitudinal studies should be conducted in the future to investigate the internal mechanisms of the factors that influence the mental health of college athletes. Moreover, this study relied on self-reported data. Additional measures that provide objective indicators of an athlete's mental state should be included in future studies. Thirdly, the sportsman in this study were all from mainland China, which may limit the generalizability of the findings. We have added the relevant information in the manuscript, please check it, thank you (please see p.10, lines 350-357).

Again we would like to thank the reviewer for the insightful comments. We think the manuscript is greatly improved and look forward to a continued positive engagement in the review process.

Reviewer 2 Report

Line No.

69-73 - Needs to be cited

89-91 -Should be cited

94 - The phrase "due to" refers to money and dates, so it is better to use "because of" ...

134 - After illustrating the figure, there should be a summary of what the figure depicts.  The introduction to the figure (required) is good but also need summarized findings in tables and figures.

147 - should explain false questionnaires.

163 - ... concise and rigorously...

207 - As indicated previously, write a short introduction before tables with a summary of findings after the tables.

265 - Do not use contractions in formal writing.

293 - ... people were advised to avoid ...

Author Response

Response to Reviewer 2 Comments

Manuscript ID: behavsci-2452211

Title: Posttraumatic Stress Disorder and Mental Health in Athletes: The Multiple Mediating Effects of Basic Psychological Needs and Generalized Anxiety Disorder

Dear Reviewer,

Thank you for the opportunity to improve our manuscript. Those comments are all valuable and very helpful for revising and improving our manuscript, as well as the important guiding significance to our research. We have studied these comments carefully and have made corrections which we hope meet with approval. We have responded to each comment and the revisions are marked in red in the manuscript. We are grateful for your positive and constructive comments and the main corrections in the manuscript and the reply to the reviewer’ s comments are as flows:

Point 1:

Line No.

69-73 - Needs to be cited

Response: Thank you for pointing out this. We have cited two references in the manuscript, thank you for your advice (please see p.6, line 76 and line 79; p.11, lines 435-438). The following are the sentences after citing the references:

These needs include the needs for competence, relatedness, and autonomy [20]. According to the self-determination theory, basic psychological needs are the driving forces behind environmental factors that have an impact on individual behavior and are necessary for healthy individual development and the realization of optimal functions [21].

  1. Viksi, A.; Tilga, H. Perceived Physical Education Teachers' Controlling Behaviour and Students' Physical Activity during Leisure Time-The Dark Side of the Trans-Contextual Model of Motivation. Sci.2022, 12, doi:10.3390/bs12090342.
  2. Schuttpelz-Brauns, K.; Hecht, M.; Hardt, K.; Karay, Y.; Zupanic, M.; Kammer, J.E. Institutional strategies related to test-taking behavior in low stakes assessment. Health Sci. Educ.2020, 25, 321-335, doi:10.1007/s10459-019-09928-y.

Point 2:

89-91 -Should be cited

Response: Thank you for pointing out this. We have cited the reference in the manuscript, thank you for your advice (please see p.3, line 97; p.13, lines 432-454). The following is the sentence after citing the reference:

The hallmark of generalized anxiety disorder, which is a common chronic mental illness among college students, is excessive, hard-to-control anxiety that causes considerable suffering and impairment [28].

  1. Kanuri, N.; Taylor, C.B.; Cohen, J.M.; Newman, M.G. Classification models for subthreshold generalized anxiety disorder in a college population: Implications for prevention. Anxiety Disord.2015, 34, 43-52, doi:https://doi.org/10.1016/j.janxdis.2015.05.011.

Point 3:

94 - The phrase "due to" refers to money and dates, so it is better to use "because of"

Response: We thank the reviewer for your suggestion. We have changed due to to because of  in the manuscript (p.3, line 98).

Point 4:

134 - After illustrating the figure, there should be a summary of what the figure depicts. The introduction to the figure (required) is good but also need summarized findings in tables and figures.

Response:

We appreciate your comments and thank you for your suggestions. In order to make the figure more clearly explained, we updated the picture and further explained it. In our paper, summary of what the figure depicts is: Fig. 1 illustrates three mediation paths: β1–β6, β5–β3, and β1–β2–β3. In addition, we mentioned in the footnote below figure 1: The hypothetical chain mediating effect model of basic psychological needs and generalized anxiety disorder between PSTD  and mental health among college athletes (Please see p.3, lines 132-133 and figure 1).

Figure 1 is depicted in the attachment.

Fig. 1. The hypothetical chain mediating effect model of basic psychological needs and generalized anxiety disorder between PSTD and mental health among college athletes.

Point 5:

147 - should explain false questionnaires.

Response: Thanks for your reminding. We apologize for the lack of clarity of expression in the manuscript. In fact, that refers to the invalid questionnaire. Questionnaires that were incomplete and had the same answer for all questions were identified as invalid questionnaires and excluded. A total of 665 subjects were included after excluding invalid questionnaires, with an effective recovery rate of 95.8%. In addition, we have change false questionnaires to invalid questionnaires (p.4, lines 155-158).

Point 6:

163 - ... concise and rigorously...

Response: We would like to thank the reviewer for your careful reading and useful suggestions. Aaccording to your suggestion, we have change concise or rigorously to concise and rigorously(p.5, line 174), thank you very much.

Point 7:

207 - As indicated previously, write a short introduction before tables with a summary of findings after the tables.

Response: Thank you very much for your careful examination and useful suggestions, we think your suggestions will be very helpful in improving our manuscript, and we have written a short introduction before tables with a summary of findings after the tables. We believe that structural changes to the results section make this section look more reasonable. Please see the results part in our manuscript, thank you.

Point 8:

265 - Do not use contractions in formal writing.

Response: Thank you for pointing out this. We have change etc to etcetera (p.8, line 286). In addition, we examined similar errors in the manuscript and corrected them.

Point 9:

293 - ... people were advised to avoid ...

Response: Thank you very much for your careful examination, we have change people are advised to avoid to people were advised to avoid (p.9, line 314).

Again we would like to thank the reviewer for the insightful comments. We think the manuscript is greatly improved and look forward to a continued positive engagement in the review process.

Reviewer 3 Report

The topic and methods are accepted by and large. Some comments to improve the manuscript:

1. Place of the study should be added to the title.

2. Inclusion criteria should be described in the abstract in more details.

3. Statistical main data should be added to the abstract.

4. It should be specified in the title and abstract that the participants were college athletes.

5. Lines 60-4 contain non-scientific sentences without any reference and overgeneralized fact. It should be cited and softened.

6. Introduction should be summarized and hypotheses should be removed. In addition, Fig-1 is not helpful enough.

7. The manuscript needs exact proof-reading.

8. In the text, the authors have emphasized on the impact of COVID pandemic on the items, although it is not reflected in the title and Abstract. It should be consistent.

The manuscript needs exact proof-reading, more specifically considering capitals.

Author Response

Response to Reviewer 3 Comments

Manuscript ID: behavsci-2452211

Title: Posttraumatic Stress Disorder and Mental Health in Athletes: The Multiple Mediating Effects of Basic Psychological Needs and Generalized Anxiety Disorder

Dear Reviewer,

Thank you for the opportunity to improve our manuscript. Those comments are all valuable and very helpful for revising and improving our manuscript, as well as the important guiding significance to our research. We have studied these comments carefully and have made corrections which we hope meet with approval. We have responded to each comment and the revisions are marked in red in the manuscript. We are grateful for your positive and constructive comments and the main corrections in the manuscript and the reply to the reviewer’ s comments are as flows:

-The topic and methods are accepted by and large. Some comments to improve the manuscript.

Response: We appreciate your comments.

Point 1:

Place of the study should be added to the title.

Response: We thank the reviewers suggestion. This study was conducted in China, and our subjects were college athletes from across China, in the midst of the COVID-19 pandemic outbreak, and we believe it has implications for similar public health events during and after the pandemic. There fore, we changed the title to  Posttraumatic Stress Disorder and Mental Health in Chinese College Athletes during and after the COVID-19 Pandemic: The Multiple Mediating Effects of Basic Psychological Needs and Generalized Anxiety Disorder. Please check the title of our manuscript. Thank you.

Point 2:

Inclusion criteria should be described in the abstract in more details.

Response: Thank the reviewer for the suggestion. Chinese college athletes who were willing to participate in the study, had participated in national competitions and had a Chinese athlete rating certificate were included in this study. We have added this information in the abstract part, thank you for your advice (please see p.1, lines 13-15).

Point 3:

Statistical main data should be added to the abstract.

Response: We thank the reviewer for your suggestion. We are sorry that we didnt added the statistical main data to the abstract. The study involved 665 college athletes (415 males and 250 females), with an average age of 20.43 years (SD = 1.68). We performed descriptive statistics, correlation analyses, and moderated chain mediation analysis using SPSS 22.0 and Hayes' PROCESS macro. The results of the final model showed that basic psychological needs were positively correlated with mental health (r = 0.443, p < 0.001), while PTSD (r = −0.346, p < 0.001) and generalized anxiety disorder (r = −0.527, p < 0.001) were negatively correlated with mental health among college athletes. There were significant indirect impacts found. According to the bootstrapping results, basic psychological needs and generalized anxiety disorder played a mediating role of 22.54% and 50.29% of the total effects of PTSD on mental health, respectively. Meanwhile, the chain mediating effect of basic psychological needs and generalized anxiety disorder (7.23%) was also significant. We have added this information in the abstract part, thank you for your advice (please see p.1, lines 17-27).

Point 4:

It should be specified in the title and abstract that the participants were college athletes.

Response: Thank you for your helpful comments and your positive review. Aaccording to your suggestion, we changed the title to  Posttraumatic Stress Disorder and Mental Health in Chinese College Athletes during and after the COVID-19 Pandemic: The Multiple Mediating Effects of Basic Psychological Needs and Generalized Anxiety Disorder. Please check the title of our manuscript. Thank you.

Point 5:

Lines 60-4 contain non-scientific sentences without any reference and overgeneralized fact. It should be cited and softened.

Response: Thanks for pointing out this. We have revised these sentences and cited references. College students (including college athletes) are currently in a critical stage of transition from adolescence to adulthood and healthy physical and mental development [17]. According to previous study, college students are more vulnerable to various effects of sudden changes in COVID-19 pandemic, and these persistent COVID-19 pandemic-related stressors may adversely affect their mental health [18]. Please see p.2, lines 66-71; p.11, lines 426-431.

  1. Zhai, S.; Qu, Y.; Zhang, D.; Li, T.; Xie, Y.; Wu, X.; Zou, L.; Yang, Y.; Tao, F.; Tao, S. Depressive symptoms predict longitudinal changes of chronic inflammation at the transition to adulthood. Immunol.2022, 13, 1036739, doi:10.3389/fimmu.2022.1036739.
  2. Zhang, M.; Qin, L.; Zhang, D.; Tao, M.; Han, K.; Chi, C.; Zhang, Z.; Tao, X.; Liu, H. Prevalence and factors associated with insomnia among medical students in China during the COVID-19 pandemic: characterization and associated factors. BMC Psychiatry2023, 23, 140, doi:10.1186/s12888-023-04556-8.

Point 6:

Introduction should be summarized and hypotheses should be removed. In addition, Fig-1 is not helpful enough.

Response: Thank you for your suggestions. In summary, athletes mental health during and after the COVID-19 pandemic and post-pandemic period is a cause for concern. Many studies have verified the negative impact of PTSD on individuals mental health and the impact of basic psychological needs and generalized anxiety disorder in promoting mental health status. We have made additions to the manuscript (please see p.3, lines 113-116).

Moreover, in order to make the figure more clearly explained, we updated the picture and further explained it. In our paper, summary of what the figure depicts is: Fig. 1 illustrates three mediation paths: β1–β6, β5–β3, and β1–β2–β3. In addition, we mentioned in the footnote below figure 1: The hypothetical chain mediating effect model of basic psychological needs and generalized anxiety disorder between PSTD and mental health among college athletes (Please see p.3, lines 132-133 and figure 1).

Figure 1 is depicted in the attachment.

Fig. 1. The hypothetical chain mediating effect model of basic psychological needs and generalized anxiety disorder between PSTD and mental health among college athletes.

Point 7:

The manuscript needs exact proof-reading.

Response: Thank you very much for your careful examination, we have enlisted the help of two graduate students majoring in English in China to polish the manuscript, and we believe that the current version of the manuscript is more readable. In addition, we have carefully proofread the full text.

Point 8:

In the text, the authors have emphasized on the impact of COVID pandemic on the items, although it is not reflected in the title and Abstract. It should be consistent.

Response: Thank you very much for your useful suggestion, we changed the title to  Posttraumatic Stress Disorder and Mental Health in Chinese College Athletes during and after the COVID-19 Pandemic: The Multiple Mediating Effects of Basic Psychological Needs and Generalized Anxiety Disorder. Please check the title of our manuscript. Thank you.

Again we would like to thank the reviewer for the insightful comments. We think the manuscript is greatly improved and look forward to a continued positive engagement in the review process.

Round 2

Reviewer 1 Report

Accept

Reviewer 3 Report

Thank you for your revisions. Please make correction on a few 'PSTD' in the text.